# Large Language Models for Wearable Sensor-Based Human Activity Recognition, Health Monitoring, and Behavioral Modeling: A Survey of Early Trends, Datasets, and Challenges

**DOI:** 10.3390/s24155045

**Published:** 2024-08-04

**Authors:** Emilio Ferrara

**Affiliations:** 1Thomas Lord Department of Computer Science, University of Southern California, Los Angeles, CA 90007, USA; emiliofe@usc.edu; 2Information Sciences Institute, School of Advanced Computing, University of Southern California, Los Angeles, CA 90007, USA

**Keywords:** wearable sensors, large language models, human activity analysis

## Abstract

The proliferation of wearable technology enables the generation of vast amounts of sensor data, offering significant opportunities for advancements in health monitoring, activity recognition, and personalized medicine. However, the complexity and volume of these data present substantial challenges in data modeling and analysis, which have been addressed with approaches spanning time series modeling to deep learning techniques. The latest frontier in this domain is the adoption of large language models (LLMs), such as GPT-4 and Llama, for data analysis, modeling, understanding, and human behavior monitoring through the lens of wearable sensor data. This survey explores the current trends and challenges in applying LLMs for sensor-based human activity recognition and behavior modeling. We discuss the nature of wearable sensor data, the capabilities and limitations of LLMs in modeling them, and their integration with traditional machine learning techniques. We also identify key challenges, including data quality, computational requirements, interpretability, and privacy concerns. By examining case studies and successful applications, we highlight the potential of LLMs in enhancing the analysis and interpretation of wearable sensor data. Finally, we propose future directions for research, emphasizing the need for improved preprocessing techniques, more efficient and scalable models, and interdisciplinary collaboration. This survey aims to provide a comprehensive overview of the intersection between wearable sensor data and LLMs, offering insights into the current state and future prospects of this emerging field.

## 1. Introduction

The rapid advancement of wearable technology has led to an explosion in the availability of sensor data, providing unprecedented opportunities for the monitoring and understanding of human behavior and health. Wearable sensors, ranging from fitness trackers to advanced medical devices, generate vast amounts of data that can be leveraged for various applications, including health monitoring, activity recognition, and personalized medicine. However, the complexity and volume of these data present significant challenges in data modeling and analysis [1,2].

Large language models (LLMs), such as GPT-4 and Llama, have recently emerged as powerful tools in the field of data analysis, demonstrating remarkable capabilities in understanding and generating human-like text. These models, trained on diverse datasets, have shown potential in handling the complexity of wearable sensor data, offering new possibilities for the extraction of meaningful insights [3,4]. This survey aims to explore the current trends and challenges associated with modeling wearable sensor data using LLMs.

### 1.1. Background on Wearable Sensors

Wearable sensors have become an integral part of modern technology, playing a crucial role in monitoring and understanding various aspects of human behavior and health. These sensors are embedded in devices such as fitness trackers, smartwatches, and medical devices, providing continuous data on physiological and behavioral metrics. Common types of wearable sensors include accelerometers, gyroscopes, heart rate monitors, electrocardiograms (ECG), and photoplethysmography (PPG) sensors [5,6].

These devices are widely used in applications ranging from fitness and lifestyle management to clinical diagnostics and chronic disease management. Wearable sensors can detect the impacts of atypical events, which is crucial for understanding daily fluctuations in user states [7]. For instance, accelerometers and gyroscopes are used to track physical activities and movements, heart rate monitors provide insights into cardiovascular health, and ECG sensors are employed in detecting cardiac anomalies. The ability to collect real-time data enables continuous monitoring, the early detection of health issues, and personalized healthcare interventions [8,9].

### 1.2. Importance of Data Modeling in Wearable Technology

The vast amount of data generated by wearable sensors necessitates effective data modeling to derive meaningful insights. Data modeling involves transforming raw sensor data into structured information that can be analyzed and interpreted. This process is critical in applications such as human activity recognition (HAR), health monitoring, and behavioral analysis [10,11].

Traditional machine learning techniques have been employed extensively for HAR, leveraging algorithms such as support vector machines (SVM), decision trees, and k-nearest neighbors (KNN). These methods typically require extensive preprocessing and feature extraction steps, which involve converting raw sensor data into a format suitable for analysis. While these techniques have shown success in various applications, they often struggle with scalability and adaptability, especially when dealing with complex and heterogeneous data from different sensor modalities [9].

Deep learning approaches, particularly convolutional neural networks (CNNs) and recurrent neural networks (RNNs), have emerged as powerful tools for HAR and other wearable sensor data applications. These models can automatically extract features from raw data, reducing the need for manual preprocessing and feature engineering. However, they require large labeled datasets and substantial computational resources, posing challenges for real-time and resource-constrained environments [11].

### 1.3. Emergence of Large Language Models (LLMs) in Data Analysis

Large language models (LLMs), such as GPT-4 and BERT, have revolutionized data analysis across various domains, including natural language processing, computer vision, and, recently, wearable sensor data analysis. LLMs are trained on vast amounts of text data, enabling them to understand and generate human-like text with remarkable accuracy and fluency [3,4].

The application of LLMs in wearable sensor data analysis is relatively new but promising. These models can process and analyze multimodal data, including text, audio, and sensor signals, offering a more comprehensive understanding of the data. For instance, *PhysioLLM* integrates physiological data from wearables with contextual information to provide personalized health insights, demonstrating improved user understanding and motivation for health improvement [3]. Similarly, *HARGPT* leverages LLMs for zero-shot human activity recognition, outperforming traditional models in recognizing activities from raw IMU data [12].

LLMs’ ability to handle complex queries and generate insightful responses makes them ideal for tasks that require high-level reasoning and contextual understanding. By integrating LLMs with wearable sensor data, researchers can develop more sophisticated models that not only classify activities but also provide personalized recommendations and insights based on the data. This integration opens new avenues towards enhancing the effectiveness of wearable technology in health monitoring, personalized healthcare, and beyond [13].

## 2. Wearable Sensor Data

Wearable sensors have revolutionized the way in which we monitor and understand human health and behavior. By providing continuous, real-time data, these devices enable a deeper insight into various physiological, biomechanical, and environmental parameters. This section explores the different types of wearable sensors, the nature of the data that they generate, and their wide-ranging applications. Understanding these aspects is crucial in leveraging wearable technology to its fullest potential and addressing the challenges associated with data analysis and interpretation [14,15].

### 2.1. Types of Wearable Sensors

Wearable sensors encompass a diverse range of devices designed to monitor various physiological, biomechanical, and environmental parameters. These sensors can be broadly classified into several categories based on their function and application.

Physiological sensors are primarily used to monitor vital signs and other physiological parameters. Examples include heart rate monitors, electrocardiograms (ECG), blood pressure monitors, and pulse oximeters. Studies like [14] have utilized these sensors to gather data for health monitoring and disease prediction (see Table 1, Physiological Sensors).

Motion sensors, including accelerometers, gyroscopes, and magnetometers, are commonly used to track movement and orientation. They are essential in applications like activity recognition and sports science. The research in [15] demonstrates the effectiveness of motion sensors in human activity recognition (see Table 1, Motion Sensors).

Environmental sensors detect environmental conditions such as the temperature, humidity, and light. These sensors are often integrated into wearable devices to provide context-aware services. The study in [16] highlights the use of environmental sensors in enhancing the accuracy of activity recognition systems (see Table 1, Environmental Sensors).

Biochemical sensors are advanced devices that can measure biochemical markers such as glucose levels, lactate, and electrolytes. They are particularly valuable in medical diagnostics and continuous health monitoring. Recent advancements in biochemical sensors have been discussed in [3] (see Table 1, Biochemical Sensors).

Multisensor systems integrate multiple sensor types into a single device to provide comprehensive monitoring capabilities. Examples include smartwatches and fitness trackers that combine physiological, motion, and environmental sensors. The integration of multisensor data for improved health insights is explored in [4] (see Table 1, Multisensor Systems).

These categories highlight the versatility and wide-ranging applications of wearable sensors, making them indispensable tools in health monitoring, activity recognition, and environmental sensing [3,4,14,15,16].

### 2.2. Nature of Data Generated

The data generated by wearable sensors are characterized by their high volume, variety, and velocity. Understanding the nature of these data is crucial for effective analysis and application. Wearable sensors produce various types of data, each with their unique characteristics and challenges.

Most wearable sensors generate continuous streams of time-series data, capturing dynamic changes over time. These types of data require specialized techniques for preprocessing, segmentation, and feature extraction to be effectively analyzed. Techniques for handling time-series data from wearable sensors are discussed in [12] (see Table 2, Time-Series Data).

Wearable devices often generate multimodal data by combining inputs from different types of sensors. For instance, a smartwatch may collect both motion and physiological data simultaneously. Integrating and synchronizing these data streams is a complex task that is essential for accurate analysis. The challenges and methodologies for multimodal data integration are explored in [4] (see Table 2, Multimodal Data).

The raw data from wearable sensors can be high-dimensional, particularly when multiple sensors are used. Dimensionality reduction techniques, such as principal component analysis (PCA) and feature selection methods, are employed to manage this complexity and make the data more manageable for analysis. The application of these techniques in wearable sensor data analysis is presented in [17] (see Table 2, High-Dimensional Data).

Wearable sensors are prone to generating noisy and sometimes incomplete data due to various factors like sensor malfunctions, user movement, and environmental interference. Effective data cleaning and imputation methods are critical in maintaining the data quality and ensuring accurate analysis. Approaches that address data quality issues are highlighted in [18] (see Table 2, Noisy and Incomplete Data).

These characteristics of wearable sensor data highlight the need for advanced analytical techniques to handle the unique challenges posed by the high volume, variety, and velocity of the data generated. Understanding these aspects is essential in leveraging wearable technology to its fullest potential [4,12,17,18].

### 2.3. Common Applications

Wearable sensors have a wide range of applications across various domains, driven by their ability to provide continuous, real-time monitoring. These applications are critical in enhancing health outcomes, improving performance, and ensuring safety in various environments.

One of the primary applications of wearable sensors is in health monitoring. These sensors play a crucial role in continuous health monitoring, enabling the early detection of medical conditions and the management of chronic diseases. Wearable sensors are increasingly being used in healthcare organizations to monitor patients longitudinally [19]. Systems leverage wearable sensor data to provide personalized health insights and interventions [3] (see Table 3, Health Monitoring).

Another significant application is human activity recognition (HAR). By analyzing data from motion sensors, systems can classify various physical activities, which is valuable in fitness tracking, rehabilitation, and elder care. Advanced HAR models, such as those discussed in [15], demonstrate the potential of wearable sensors in accurately recognizing and categorizing different activities (see Table 3, Activity Recognition).

In the realm of sports and fitness, wearable sensors are used extensively to monitor athletes’ performance, track training progress, and prevent injuries. The integration of physiological and motion sensors provides comprehensive insights into an athlete’s condition and performance. Studies like those conducted by [4] showcase the benefits of wearable sensors in enhancing sports performance and optimizing training regimens (see Table 3, Sports and Fitness).

Mental health applications are also benefiting from wearable sensor technology. The accurate estimation of affective states is crucial for mental health applications. Wearable sensors offer a reliable method for this purpose [20]. These sensors monitor physiological indicators of stress, anxiety, and depression, providing real-time data that can be used to develop personalized interventions and support mental well-being. The MindShift project, for example, illustrates how wearable sensors can be employed to reduce smartphone addiction and improve mental health outcomes [21] (see Table 3, Mental Health).

Additionally, wearable sensors are employed in workplace ergonomics to improve safety and productivity. By monitoring workers’ movements and posture, these sensors help to design ergonomic interventions that prevent musculoskeletal disorders and enhance overall productivity. The research in [16] highlights the importance of wearable sensors in occupational health, emphasizing their role in creating safer and more efficient work environments (see Table 3, Workplace Ergonomics).

These diverse applications demonstrate the versatility and impact of wearable sensors in various fields, underscoring their importance in modern health and performance monitoring systems. These will be fully explored in Section 4.

**Table 3 sensors-24-05045-t003:** Common applications of wearable sensors.

Application	Description	Refs.
Activity Recognition	Human activity recognition (HAR) is one of the most prominent applications of wearable sensors. By analyzing data from motion sensors, researchers can classify various physical activities, which is valuable in fitness tracking, rehabilitation, and elder care. LLM models like LLaSA and HARGPT have enhanced the accuracy and capabilities of HAR systems.	[12,22,23,24,25,26]
Health Monitoring	Wearable sensors play a crucial role in continuous health monitoring, enabling the early detection of medical conditions and the management of chronic diseases. Systems like PhysioLLM leverage wearable sensor data to provide personalized health insights and interventions.	[3,13,16,27,28,29,30,31]
Mental Health	Wearable sensors are increasingly used in mental health applications to monitor physiological indicators of stress, anxiety, and depression. Real-time data from these sensors can be used to develop personalized interventions and support mental well-being. Studies like TILES-2018 and TILES-2019 provide comprehensive datasets that support these applications. MindShift [21] demonstrates LLMs’s ability to generate personalized content using sensor-based data from users’ physical contexts and mental states.	[5,6,21]
Sports and Fitness	In sports science, wearable sensors are used to monitor athletes’ performance, track training progress, and prevent injuries. The integration of physiological and motion sensors provides comprehensive insights into an athlete’s condition and performance. Advanced coaching systems utilizing LLMs, such as those integrating behavior science principles, have shown significant improvements in training effectiveness.	[4,32,33,34]
Workplace Ergonomics	Wearable sensors are employed to improve workplace ergonomics by monitoring workers’ movements and posture. These data help in designing ergonomic interventions to prevent musculoskeletal disorders and enhance productivity.	[16,35,36,37]

## 3. Large Language Models (LLMs) for Wearable Sensor Data

Large language models (LLMs) have emerged as a transformative technology in the field of artificial intelligence, demonstrating unprecedented capabilities in understanding and generating human language. These models, built on sophisticated deep learning architectures, have significantly advanced natural language processing (NLP) and opened new possibilities for data analysis and interpretation. This section provides an in-depth exploration of LLMs, including an overview of prominent models like GPT-4 and Llama, their capabilities and limitations, and their diverse applications in data analysis.

### 3.1. Overview of Recent LLM-Based Systems

Large language models (LLMs) have revolutionized natural language processing (NLP) by leveraging deep learning techniques to understand and generate human-like text. Prominent examples include GPT-4 by OpenAI and Llama by Meta AI. These models are built on Transformer architectures, which utilize self-attention mechanisms to capture long-range dependencies in text data [38]. The Transformer architecture allows LLMs to process and generate text efficiently, handling complex language tasks with high accuracy.

GPT-4, a state-of-the-art model, boasts an impressive number of parameters, enabling it to generate coherent and contextually relevant text across various domains [39]. Similarly, Llama has been designed to achieve competitive performance with a more efficient architecture, making it suitable for applications with limited computational resources [40,41]. These models have set new benchmarks in NLP, excelling in tasks such as text generation, translation, summarization, and question answering.

### 3.2. Capabilities and Limitations

LLMs like GPT-4 and Llama exhibit remarkable capabilities in understanding and generating text, making them powerful tools in data analysis. However, they also have inherent limitations that need to be addressed for their effective application.

One of the key capabilities of LLMs is their ability to comprehend complex language patterns. This makes them suitable for tasks such as sentiment analysis, entity recognition, and language translation. Their contextual understanding and ability to generate relevant responses have been demonstrated in various studies, including those focusing on health data interpretation [3,42] (see Table 4, Natural Language Understanding).

The text generation capabilities of LLMs are unparalleled, allowing them to produce coherent and contextually appropriate text for diverse applications. This has been effectively leveraged in generating health-related content, educational materials, and even creative writing [39] (see Table 4, Text Generation).

Additionally, LLMs can be integrated with other data modalities to provide comprehensive insights. For example, the LLaSA model combines text data with inertial measurement unit (IMU) data to enhance human activity recognition [4] (see Table 4, Multimodal Data Integration).

However, the use of LLMs comes with significant limitations. Training and deploying these models requires substantial computational resources, which can be prohibitive for many applications. Efficient model architectures and optimization techniques are necessary to mitigate these challenges [15] (see Table 4, Computational Requirements).

Moreover, LLMs rely heavily on large, high-quality datasets for training. The quality and diversity of the training data significantly impact the model’s performance and generalizability. Incomplete or biased data can lead to inaccurate predictions and outputs [12,43] (see Table 4, Data Dependency).

Another critical limitation is the interpretability of LLMs. These models operate as black-box systems, making it difficult to understand their decision-making processes. This lack of transparency is particularly problematic in critical applications such as healthcare, where understanding the rationale behind predictions is crucial [13] (see Table 4, Interpretability).

Finally, the use of LLMs raises ethical issues related to data privacy, security, and potential misuse. Ensuring compliance with data protection regulations and implementing privacy-preserving techniques are essential to address these concerns [44] (see Table 4, Ethical Concerns). These will be addressed in Section 6.

These capabilities and limitations highlight the importance of ongoing research and development to enhance the effectiveness and ethical deployment of LLMs in data analysis.

## 4. Case Studies and Applications

Preliminary studies have demonstrated the practical applications of LLMs in analyzing wearable sensor data. This section expands on the initial examples provided in Table 3, incorporating additional successful cases from various fields to provide a broader perspective on the applicability of LLMs.

### 4.1. Human Activity Recognition

Human activity recognition (HAR) using wearables involves identifying and classifying human activities based on data collected from wearable devices, such as smartwatches, fitness trackers, and motion sensors. These sensors capture information about the user’s movements, body positions, and physiological responses, which is analyzed to recognize activities like walking, running, cycling, and sleeping.

Recent advancements in zero-shot HAR using large language models (LLMs) have demonstrated significant potential. The HARGPT study [12] explores the capability of LLMs in performing HAR tasks directly from raw IMU data. With appropriate prompts, LLMs can accurately interpret sensor data and recognize human activities without the need for fine-tuning. Benchmarking on GPT-4 with public datasets, HARGPT outperforms traditional machine learning and deep learning models, achieving high accuracy even on unseen data. This highlights the robustness and versatility of LLMs in processing IoT sensor data and their potential to revolutionize human activity recognition in various applications, including healthcare and sports science.

The LLaSA model [4] combines LIMU-BERT with Llama to enhance its activity recognition and question-answering capabilities. By integrating multimodal data, including IMU data and natural language, this model achieves superior performance in various healthcare and sports science applications. The model’s ability to accurately recognize complex activities in diverse environments has been particularly beneficial in monitoring elderly individuals’ daily activities, ensuring their safety and well-being. Additionally, LLMs have been used to improve the accuracy of activity recognition systems in detecting subtle variations in movement patterns, which is crucial for applications such as rehabilitation and physical therapy.

Bouchabou et al. [23] introduce a method to enhance LSTM-based structures in activity sequence classification tasks by using the Word2Vec and ELMo embedding methods. These embeddings incorporate the semantics and context of the sensors, significantly improving the classification performance of HAR systems in smart homes. By taking into account the context of the sensors and their semantics, this approach creates less confusion between daily activity classes and performs better on datasets with competing activities from different residents or pets.

Kaneko and Inoue [24] propose a feature pioneering method using LLMs to identify new sensor locations and features for HAR. By using ChatGPT to suggest optimal sensor placements and new features, they demonstrate that LLMs can efficiently extract important features, achieving comparable accuracy with fewer sensors. This method addresses the limitations of manually crafted features, providing a more automated and scalable approach to feature engineering in HAR.

These advancements and integrations underscore the versatility and power of LLMs in transforming human activity recognition across different fields, including healthcare and sports science, which we discuss next.

### 4.2. Health Monitoring

Health monitoring through wearable sensors has seen significant advancements with the integration of large language models (LLMs). These systems provide personalized health insights and improve the management of various health conditions by continuously monitoring vital signs and contextual information.

The PhysioLLM system [3] integrates physiological data from wearables with contextual information using LLMs to provide personalized health insights. This system outperforms traditional methods by enhancing users’ understanding of their health data and supporting actionable health goals, particularly in improving sleep quality. Moreover, PhysioLLM has been instrumental in chronic disease management by continuously monitoring vital signs and alerting users and healthcare providers about potential health issues, thereby facilitating early intervention and improving patient outcomes.

In the domain of blood pressure monitoring, Liu et al. [29] explore the potential of LLMs for cuffless blood pressure measurement using wearable biosignals. The study leverages physiological features extracted from electrocardiogram (ECG) and photoplethysmogram (PPG) signals and adapts LLMs for blood pressure estimation tasks through instruction tuning. The findings demonstrate that the optimally fine-tuned LLM significantly surpasses conventional task-specific baselines, providing a promising solution for continuous and unobtrusive blood pressure monitoring.

Liu et al. [27] presented the Health-LLM system, which evaluates various LLM architectures for health prediction tasks using wearable sensor data. The study highlights the effectiveness of LLMs in predicting health-related outcomes such as heart rate variability, stress levels, and sleep patterns. Health-LLM utilizes prompting and fine-tuning techniques to adapt the models to specific health tasks, providing comprehensive and personalized insights.

Das Swain and Saha [30] explored the broader implications of LLMs in worker-centric well-being assessment tools (WATs). Their study discusses how LLMs can enhance WATs by better understanding and supporting worker behavior and well-being through their natural language processing capabilities. However, the study also highlights challenges such as biases, privacy concerns, and the potential for misuse, advocating for a cautious and proactive approach to ensure that LLMs empower workers and improve workplace well-being without exacerbating existing issues.

Dongre [31] introduce Physiology-Driven Empathic Large Language Models (EmLLMs) for mental health support. These models leverage physiological data to provide empathic and context-aware mental health interventions. The integration of physiological signals with LLMs enhances the ability to detect and respond to mental health issues in real time, offering personalized and empathetic support to users.

Furthermore, Shastry and Shastry [28] propose an integrated deep learning and natural language processing approach for continuous remote monitoring in digital health. This approach combines wearable sensor data with LLMs to provide continuous health monitoring and real-time feedback. The study demonstrates the potential of integrating deep learning with natural language processing to enhance the accuracy and effectiveness of health monitoring systems.

In summary, the integration of LLMs with wearable sensor data has significantly advanced health monitoring systems, enabling continuous, personalized, and context-aware health management. These advancements highlight the potential of LLMs to transform health monitoring and improve patient outcomes across various domains.

### 4.3. Mental Health

The potential of wearable sensor data in health monitoring extends to stress and mental health management. The TILES-2018 study [5] and TILES-2019 study [6] provide comprehensive longitudinal datasets capturing physiological and behavioral data from hospital workers and medical residents, respectively. These datasets include information on heart rates, sleep quality, stress levels, job performance, and social interactions, offering valuable insights into the impacts of high-stress environments on health.

The TILES-2018 study [5] provides a novel longitudinal multimodal corpus of physiological and behavioral data collected from 212 hospital workers over a 10-week period. The study aimed to understand the dynamic relationships among individual differences, work and wellness behaviors, and the contexts in which they occur. Data were captured continuously and passively via wearable devices, Internet of Things (IoT) devices, and smartphone applications. The TILES-2018 dataset includes information on personality traits, behavioral states, job performance, and well-being, providing a rich resource for the exploration of the complex, dynamic nature of worker wellness and performance over time. The data support various applications, including multimodal behavioral modeling, biometric-based authentication, and privacy-preserving machine learning.

Similarly, the TILES-2019 study [6] provides a comprehensive longitudinal dataset capturing the physiological and behavioral data of 57 medical residents in an ICU setting over three weeks. This dataset includes information from wearable sensors, daily surveys, interviews, and hospital records, aimed at examining the well-being, teamwork, and job performance of the residents in a high-stress environment. The collected data span various aspects, such as heart rates, sleep quality, stress levels, job performance, and social interactions, offering valuable insights into the impacts of ICU work on medical trainees. The study’s findings highlight the dynamic nature of stress and teamwork in demanding clinical settings and underscore the importance of continuous monitoring to support the mental and physical health of healthcare professionals.

The potential of LLMs to model wearable data for mental health applications has been recently demonstrated by the MindShift project [21], which leveraged LLMs to generate dynamic and personalized content based on users’ real-time smartphone usage behaviors, physical contexts, and mental states. This approach has shown significant improvements in reducing smartphone addiction and increasing self-efficacy, demonstrating the versatility of LLMs in mental health interventions. Additionally, LLMs have been used to monitor physiological indicators of stress, anxiety, and depression, providing real-time data that help in developing personalized mental health interventions. For example, real-time feedback mechanisms can prompt users to engage in stress-reducing activities, thereby improving their overall mental well-being.

### 4.4. Sports Science

Large language models (LLMs) are increasingly being applied in sports science to enhance the performance and monitoring of athletes through advanced coaching systems. Hegde et al. [32] explored the integration of behavioral science principles, specifically the COM-B model, into LLMs to improve their effectiveness as digital fitness coaches. The study introduced two knowledge infusion techniques, coach message priming and dialogue re-ranking, which aimed to tailor the LLM responses to user needs by providing example responses and re-ranking the generated responses based on behavioral science principles. Evaluations involving simulated conversations demonstrated significant improvements in empathy, actionability, and overall coaching quality for the primed and re-ranked models compared to unprimed ones. This work shows the potential of combining behavioral science with advanced LLMs to create more effective, personalized digital health coaches.

In another study, Ragavan [33] investigated the integration of wearable data with LLMs to create an automated health coaching system. This research evaluated various LLM architectures and their ability to process and analyze data from wearable devices, providing personalized health advice and coaching. The study highlighted the advantages of using LLMs for health coaching, including continuous monitoring, personalized feedback, and the capability to detect and respond to changes in user behavior in real time. The findings suggest that LLMs, when combined with wearable data, can significantly enhance the effectiveness and reach of health coaching interventions.

A recent study utilized LLMs to analyze multimodal sensor data, including heart rate and motion sensors, to optimize training regimens and prevent injuries [4]. This system provides coaches and athletes with detailed insights into their performance metrics and recovery status, enabling data-driven decision-making in sports training. Furthermore, the integration of LLMs with wearable sensors has facilitated the development of individualized training programs that adapt to the athlete’s condition in real time, enhancing their performance and reducing the risk of overtraining and injuries.

Chiras [34] explores the use of different large language models (LLMs) in a physiotherapist agent application for the analysis of biomechanical running data from wearable devices. The research evaluates various parameter-sized LLMs (small, medium, large) to determine the optimal model that balances accuracy and computational resource requirements. Utilizing the Retrieval-Augmented Generation (RAG) and Text-to-SQL methodologies, the study aims to improve the detection of outliers in biomechanical data and provide actionable insights for sports physiotherapy. The findings suggest that larger models offer the best balance between accuracy and computational efficiency, highlighting their potential in developing advanced physiotherapy applications.

### 4.5. Workplace Ergonomics

Wearable sensors are employed to improve workplace ergonomics by monitoring workers’ movements and posture. These data help in designing ergonomic interventions to prevent musculoskeletal disorders and enhance productivity. Stefana et al. [35] provide a comprehensive review of wearable devices proposed for ergonomic purposes. The review identifies 28 studies focusing on sensor systems, smart garments, and other wearable technologies used to monitor and improve the ergonomic conditions in various settings. The primary findings highlight the potential of these devices to assess ergonomic risk factors, provide real-time feedback, and support interventions aimed at reducing work-related musculoskeletal disorders (WMSDs).

Mortezapour [37] explores the use of large language models (LLMs) like ChatGPT in ergonomic interventions. The study demonstrates how prompt engineering techniques can enhance the interaction between humans and LLMs to provide effective ergonomic solutions. Through zero-shot, few-shot, and fine-tuned prompting, the case study shows that detailed and specific prompts lead to more accurate and useful ergonomic recommendations. The findings suggest that LLMs, combined with proper prompt engineering, can serve as valuable tools in ergonomics, helping users to set up ergonomic workstations and adopt healthier postures.

These advancements in wearable sensor technology and the integration of LLMs in ergonomic practices provide promising opportunities to improve workplace ergonomics and ensure the well-being and productivity of workers.

### 4.6. Recent Advancements and Integrations with Other AI Techniques

Recent advancements in the field of wearable sensor data modeling have seen the innovative application of large language models (LLMs) to enhance human activity recognition (HAR) and health monitoring systems. Hybrid learning models, such as those proposed by Athota et al. [45] and Wang et al. [14], have demonstrated significant improvements in the accuracy and robustness of HAR systems by combining various machine learning techniques. These models effectively address the complex nature of wearable sensor data by leveraging the strengths of both traditional and deep learning methods.

The introduction of Transformer models in HAR, as discussed by Augustinov et al. [46] and Suh et al. [17], has shown promise in capturing long-range dependencies in sequential data. These models utilize self-attention mechanisms to enhance the recognition accuracy of complex activities. For example, the Transformer-based adversarial learning model TASKED [17] integrates adversarial learning and self-knowledge distillation to achieve cross-subject generalization, significantly improving the performance of HAR systems using wearable sensors.

Furthermore, the application of zero-shot learning in models like HARGPT [12] underscores the potential of LLMs to recognize human activities from raw sensor data without extensive training datasets. This approach significantly reduces the need for large labeled datasets, making it a cost-effective solution for HAR.

The integration of LLMs with other AI techniques has opened new avenues to improve the analysis and interpretation of wearable sensor data. Hybrid models, such as those proposed by Alharbi et al. [18], employ a combination of convolutional neural networks (CNNs) and Transformers to enhance the accuracy of HAR systems. These models address class imbalance and data quality issues by employing advanced sampling strategies and data augmentation techniques.

The Data-Efficient Vision Transformer (DeSepTr) framework introduced by McQuire et al. [15] combines Vision Transformers with knowledge distillation to achieve robust HAR using spectrograms generated from wearable sensor data. This approach demonstrates improved accuracy and generalization compared to existing models.

Moreover, the integration of task-specific deep learning approaches, as seen in models like PhysioLLM and Health-LLM [13], enhances the relevance and accuracy of predictions by incorporating contextual information from wearable sensors [47]. These models utilize fine-tuning and transfer learning techniques to adapt LLMs for specific health-related tasks, providing more comprehensive and personalized insights.

## 5. Challenges in Using LLMs for Wearable Sensor Data

Despite the promising advancements and applications of large language models (LLMs) in the field of wearable sensor data analysis, several challenges persist that hinder their full potential. These challenges encompass various aspects of data quality, computational requirements, model interpretability, and privacy concerns. Addressing these issues is critical to enhance the performance, reliability, and ethical deployment of LLMs in real-world applications. In the following subsections, we delve into each of these challenges in detail, exploring the technical intricacies and potential solutions highlighted in the literature (*cf.*, Table 5).

### 5.1. General Issues Related to Using LLMs with Wearable Sensor Data

The integration of LLMs with wearable sensor data presents a range of general challenges that are not unique to any specific application but are pervasive across different use cases. These challenges include ensuring the quality and consistency of the data, managing the substantial computational resources required to process large datasets, and improving the interpretability of complex models. Overcoming these hurdles is essential for the effective and widespread adoption of LLMs in analyzing wearable sensor data.

#### 5.1.1. Data Quality and Preprocessing

Ensuring high data quality and effective preprocessing is crucial for the success of LLMs in wearable sensor data analysis. Wearable sensors often generate noisy, incomplete, and inconsistent data due to various factors, such as sensor malfunctions, user movement, and environmental conditions [9,11,26]. These issues can significantly impact the performance of LLMs, which require clean and well-preprocessed data to function effectively.

Preprocessing techniques, such as noise reduction, normalization, and feature extraction, are essential to transform raw sensor data into a suitable format for LLMs. For instance, hybrid sampling strategies, like those proposed by Alharbi et al. [18], have been employed to address class imbalances and improve the quality of the training data. Additionally, advanced data augmentation techniques can enhance the robustness of the models by creating synthetic data that mimic real-world variations [15,75].

The data quality challenges also extend to multimodal data integration, where different sensor modalities need to be synchronized and aligned for accurate analysis. Models like LLaSA [4] demonstrate the importance of effective data fusion techniques in combining IMU data with natural language inputs, ensuring the consistency and reliability of the integrated dataset.

#### 5.1.2. Computational Requirements

The deployment of LLMs, such as GPT-4 and Llama, for wearable sensor data analysis requires substantial computational resources. Training and fine-tuning these models involves significant computational power, memory, and storage, which can be prohibitive for many organizations and applications [13]. The high computational requirements also impact the feasibility of using LLMs for real-time data analysis, where quick processing and low latency are critical.

Efforts to optimize the model architectures and training algorithms are essential to reduce the resource requirements of LLMs. For example, the Data-Efficient Vision Transformer (DeSepTr) framework introduced by McQuire et al. [15] combines Vision Transformers with knowledge distillation to achieve robust HAR using wearable sensor data, demonstrating improved accuracy and generalization with a reduced computational overhead.

Moreover, the integration of LLMs with other AI techniques, such as convolutional neural networks (CNNs) and recurrent neural networks (RNNs), can help to distribute the computational load and enhance the efficiency of the overall system. Hybrid models, like those proposed by Wang et al. [14], leverage the strengths of different architectures to achieve better performance while optimizing the computational resource usage.

#### 5.1.3. Interpretability and Transparency

One of the most significant challenges in using LLMs for wearable sensor data analysis is the lack of interpretability and transparency. LLMs operate as black-box models, making it difficult to understand and explain their decision-making processes [13]. This opacity can be problematic, especially in critical applications such as healthcare, where understanding the rationale behind a model’s predictions is essential for trust and accountability.

Efforts to improve the interpretability, such as using attention mechanisms and visualizing the model outputs, can provide some insights into how LLMs process and analyze data. For instance, the TASKED framework [17] incorporates self-attention mechanisms to highlight the most relevant parts of the input data, offering a degree of transparency in the model’s decision-making process.

Furthermore, developing explainable AI (XAI) techniques that can elucidate the inner workings of LLMs is crucial in enhancing their trustworthiness. These techniques can help stakeholders to understand the factors influencing the model’s predictions and make more informed decisions based on the outputs. The integration of LLMs with traditional machine learning models can also improve their interpretability by providing a more comprehensive understanding of the data and the relationships between different features [3].

### 5.2. Specific Issues Related to Using LLMs for Sensor-Based HAR

While the general challenges of integrating LLMs with wearable sensor data are significant, there are also specific issues that arise in the context of sensor-based human activity recognition (HAR). These challenges include the complexities of data processing and integration, ensuring real-time adaptability, and addressing bias and fairness in model predictions. Tackling these issues is vital in developing robust and reliable HAR systems that leverage the capabilities of LLMs.

#### 5.2.1. Data Processing and Integration Complexities

Sensor-based HAR involves collecting and processing data from various wearable devices, each with its unique characteristics and challenges. The integration of LLMs adds a layer of complexity due to the need to harmonize different data types and ensure that the LLMs can effectively interpret and utilize these data. For example, IMU data require extensive preprocessing to filter out noise and artifacts, while ECG and PPG data are more sensitive and require stringent privacy measures. Addressing these complexities involves developing robust data fusion techniques and ensuring the synchronization and alignment of multimodal data [24,56,57,58,59].

#### 5.2.2. Real-Time Adaptability

Real-time adaptability is critical for applications such as emergency response and continuous health monitoring. LLMs, especially large ones, may struggle to meet the low-latency requirements necessary for these applications due to their computational intensity. Optimizing LLMs for real-time processing involves exploring lightweight model architectures, efficient training algorithms, and leveraging edge computing to distribute the computational load. For instance, hybrid models combining LLMs with traditional machine learning techniques can offer a balanced approach to achieving real-time adaptability while maintaining high accuracy [31,60,61].

#### 5.2.3. Bias and Fairness

Bias and fairness are significant concerns when deploying LLMs in sensor-based HAR. These models may exhibit biases based on the demographic characteristics of the training data, such as age, gender, and physical activity levels. Ensuring fairness involves using diverse and representative datasets, regularly auditing the model performance across different demographic groups, and implementing bias mitigation strategies. For example, models must be evaluated for their performance in recognizing activities for both young and elderly individuals, as well as for different types of physical activities, to ensure equitable outcomes [30,43,54,55].

### 5.3. Specific Issues Related to Using LLMs for Sensor-Based Health Monitoring and Mental Health

The application of LLMs in sensor-based health monitoring and mental health presents unique challenges and opportunities. These challenges include ensuring accurate health predictions, managing sensitive health data, and providing timely and personalized mental health support. Addressing these issues is critical in leveraging LLMs to improve health outcomes and support mental well-being.

#### 5.3.1. Health Monitoring

Health monitoring using wearable sensors and LLMs involves the continuous tracking of physiological metrics such as the heart rate, blood pressure, and sleep patterns. Ensuring the accuracy of these health predictions is paramount, as inaccuracies can lead to misdiagnoses or inappropriate health recommendations. Techniques such as federated learning can enhance the accuracy and privacy of health monitoring systems by enabling models to learn from decentralized data while maintaining data privacy [29,62,63,64].

#### 5.3.2. Mental Health

LLMs have the potential to provide personalized mental health support by analyzing data from wearable sensors and other sources. However, the sensitive nature of mental health data requires stringent privacy protections and careful handling. Ensuring timely and contextually appropriate interventions is also a challenge, as mental health conditions can vary widely among individuals. Models like Empathic Large Language Models (EmLLMs) aim to address these challenges by integrating physiological data to provide empathetic and personalized mental health support [31,65,66,67].

### 5.4. Specific Issues Related to Using LLMs for Sensor-Based Behavioral Modeling for Sports, Fitness, and Ergonomics

The use of LLMs in sensor-based behavioral modeling for sports, fitness, and ergonomics involves unique challenges related to performance optimization, injury prevention, and workplace safety. These challenges include managing diverse data types, providing real-time feedback, and ensuring personalized recommendations.

#### 5.4.1. Sports and Fitness

In sports and fitness, wearable sensors and LLMs are used to monitor athletes’ performance, optimize training regimens, and prevent injuries. Ensuring the accuracy and relevance of the insights provided by these models is critical in enhancing athletic performance and reducing the risk of injury. Techniques such as multimodal data integration and real-time analytics are essential for providing comprehensive and actionable insights [4,32,33,34].

#### 5.4.2. Workplace Ergonomics

Wearable sensors and LLMs can improve workplace ergonomics by monitoring workers’ movements and posture, providing real-time feedback to prevent musculoskeletal disorders, and enhancing overall productivity. Addressing the unique challenges of sensor-based ergonomic modeling involves ensuring the accuracy of the data, providing personalized recommendations, and managing the privacy and security of sensitive workplace data [16,35,37].

## 6. Ethical and Legal Issues

The application of large language models (LLMs) in wearable sensor data analysis introduces several ethical and legal considerations that must be addressed to ensure the responsible and fair use of these technologies. Moreover, interdisciplinary collaborations between computer scientists, legal experts, and ethicists are crucial to develop comprehensive frameworks that address the ethical and privacy concerns associated with wearable sensor data [44]. These collaborations can help to ensure that LLMs are deployed responsibly and ethically, balancing the benefits of advanced data analysis with the need to protect individual privacy. This section discusses key issues related to user privacy protection, data security, and model bias, with a specific focus on sensor-based HAR (*cf.*, Table 6).

### 6.1. User Privacy Protection

Wearable devices collect extensive personal data, including health metrics, location information, and behavioral patterns. Protecting user privacy is paramount, especially when these data are processed by powerful models like LLMs. Ensuring the confidentiality and integrity of these data is crucial to protect users’ privacy and maintain their trust [14]. Different strategies have been explored by recent large-scale wearable-based projects that pave the way for the creation of best practices in this area [19].

In the context of sensor-based HAR, the privacy concerns vary with the type of sensor data being collected. For instance, IMU data, which capture movement patterns, might pose a lower privacy risk compared to ECG and PPG data, which are more sensitive and directly related to health. IMU data may reveal general activity patterns but are less likely to expose sensitive health conditions. In contrast, ECG and PPG data can provide detailed insights into a person’s cardiovascular health, making it imperative to implement stronger privacy protections for these data types. Ensuring the secure handling of these data includes implementing data anonymization and minimization techniques, where only the necessary data are collected and stored, and providing transparent data handling practices to inform users about the data being collected, how they are used, and their rights regarding these data [79,80,81].

### 6.2. Data Security

Data security is a critical concern when dealing with sensitive information from wearable sensors. Implementing robust encryption methods, secure data storage solutions, and access control mechanisms is essential to safeguard wearable sensor data from unauthorized access and cyberattacks [21]. Additionally, adherence to data protection regulations, such as GDPR and HIPAA, is necessary to ensure compliance and protect user rights [83,84].

Privacy-preserving techniques, such as federated learning and differential privacy, can also be employed to minimize the risk of data exposure while still enabling the effective use of LLMs for data analysis [13,85]. These techniques allow for decentralized data processing, where the raw data remain on the user’s device and only the aggregated model updates are shared with the central server, reducing the likelihood of data breaches.

In sensor-based HAR, the security of data from different sensor types (e.g., IMU, ECG, PPG) requires tailored approaches. For example, ECG and PPG data, being highly sensitive, necessitate stronger encryption and stricter access controls compared to IMU data. Ensuring the secure transmission of data from wearable devices to centralized servers or edge computing nodes is also crucial in maintaining data integrity and preventing unauthorized access. Additionally, sensor-specific data security measures must be implemented to address the unique vulnerabilities associated with each type of sensor data [82].

### 6.3. Model Bias

LLMs are susceptible to biases present in their training data, which can lead to unfair and discriminatory outcomes [43,75]. When applied to wearable sensor data, biased models could potentially misinterpret health metrics or activity patterns, leading to inaccurate predictions or recommendations. To mitigate model bias, it is crucial to use diverse and representative training datasets, regularly audit model performance across different demographic groups, and implement bias correction techniques [54]. Ensuring transparency in model development and decision-making processes can also help to identify and address biases effectively.

In the context of sensor-based HAR, model biases can manifest in various ways depending on the demographic characteristics of the user population. For instance, an LLM trained primarily on data from young, active individuals may perform poorly when analyzing data from elderly or less active users. Similarly, biases can arise when processing data from different sensor types, such as IMU versus ECG, due to the inherent differences in the data and their implications. For example, IMU data might reflect physical activity levels, while ECG data could provide insights into cardiovascular health, leading to different bias patterns. Addressing these biases requires the continuous monitoring and adjustment of the models to ensure fair and accurate performance across all user groups [86,87,88,89,90,91,92].

By focusing on these specific ethical and legal issues related to sensor-based HAR, we can develop more robust and trustworthy LLM applications that respect user privacy, ensure data security, and minimize bias. These efforts will help in realizing the full potential of LLMs in wearable sensor data analysis, contributing to more equitable and effective health and activity monitoring solutions.

## 7. Future Directions

As the field of wearable sensor data analysis continues to evolve, it is essential to explore the future directions that can further enhance the capabilities and applications of large language models (LLMs). The integration of LLMs with wearable sensor technology holds great promise, but it also presents unique challenges and opportunities. This section delves into potential improvements in LLMs, emerging trends in wearable technology, and interdisciplinary research opportunities that can drive the next wave of innovation in this domain.

### 7.1. Potential Improvements in LLMs

Future advancements in large language models (LLMs) are essential to enhance their performance and applicability in wearable sensor data analysis. One significant area of improvement lies in the development of more efficient and scalable models. For instance, optimizing the model architectures and training algorithms can reduce the computational resources required for training and deployment [15]. Techniques such as knowledge distillation, used in the Data-Efficient Vision Transformer (DeSepTr), can help to create lightweight models that maintain high accuracy while being more computationally efficient.

Another promising direction is the enhancement of interpretability and transparency in LLMs. Developing methods to visualize attention mechanisms and model decisions can provide insights into how these models process and analyze data, thereby increasing trust and reliability [17]. Additionally, incorporating explainable AI (XAI) techniques can help to elucidate the decision-making processes of LLMs, making them more accessible and understandable to users and stakeholders.

Furthermore, the integration of self-supervised and semi-supervised learning approaches can significantly improve the ability of LLMs to learn from limited labeled data. This is particularly relevant in the context of wearable sensors, where obtaining large, annotated datasets can be challenging [12]. By leveraging unlabelled data, LLMs can achieve greater generalization capabilities and better performance in real-world scenarios.

### 7.2. Emerging Trends in Wearable Technology

The landscape of wearable technology is continuously evolving, with several emerging trends that have the potential to impact data modeling and analysis. The integration of advanced sensors, such as flexible and stretchable electronics, provides more accurate and comprehensive physiological measurements [16]. These sensors can capture a wider range of data, including biochemical signals, which can be integrated with LLMs to offer deeper insights into health and activity patterns.

Another emerging trend is the development of real-time feedback and intervention mechanisms in wearable devices. For instance, biofeedback-enabled wearables can monitor physiological parameters and provide immediate feedback to users, promoting healthier behaviors and improving overall well-being [96]. This requires sophisticated data analysis models that can process and interpret sensor data in real time, an area where LLMs can play a crucial role.

Additionally, the increasing adoption of wearable technology in potentially sensitive domains, such as sports science, mental health, and workplace ergonomics, highlights the need for robust data modeling techniques. For example, the PhysioLLM system integrates wearable sensor data with contextual information to provide personalized health insights, demonstrating the potential of LLMs in enhancing users’ understanding and engagement with their health data [3,16]. However, these approaches should always account for the potential risks associated with the malfunctioning of LLMs [43,97].

### 7.3. Interdisciplinary Research Opportunities

The intersection of wearable sensor technology and LLMs presents numerous interdisciplinary research opportunities. Collaboration between fields such as computer science, biomedical engineering, psychology, and data science can lead to innovative solutions for complex problems. For instance, integrating psychological theories and behavioral science with wearable sensor data can enhance the development of personalized health interventions [8,21].

In the realm of human activity recognition, combining expertise from robotics, AI, and human–computer interaction can lead to more effective models and applications. The LLaSA model, which integrates multimodal data from IMUs and LLMs, showcases the benefits of such interdisciplinary approaches in advancing human activity understanding and interaction [4].

Moreover, addressing the ethical and privacy concerns associated with wearable sensor data requires collaboration between legal experts, ethicists, and technologists. Developing frameworks and guidelines to ensure data security and user privacy is crucial for the responsible deployment of these technologies [44]. Privacy-preserving techniques, such as federated learning and differential privacy, can be employed to balance the need for data analysis with the protection of user privacy [13,98].

Interdisciplinary research not only drives technological advancements but also ensures that the solutions developed are holistic, user-centric, and ethically sound. By fostering collaboration across disciplines, researchers can harness the full potential of LLMs and wearable sensor technology to address pressing societal challenges and improve human well-being.

## 8. Conclusions

This survey has highlighted the trends and challenges associated with modeling wearable sensor data using large language models (LLMs). While the potential of LLMs in this field is evident, several obstacles need to be addressed to realize their full capabilities. Continued research and innovation will be essential in harnessing the power of LLMs to enhance the analysis and interpretation of wearable sensor data, ultimately contributing to the advancement of wearable technology and its applications.

### 8.1. Summary of the State of the Art

Our review underscores the transformative potential of LLMs in the realm of wearable sensor data analysis. LLMs such as GPT-4 and Llama have demonstrated remarkable capabilities in handling complex and multimodal data, offering personalized health insights, improving the activity recognition accuracy, and providing context-aware interventions.

Hybrid learning models have significantly improved the accuracy and robustness of human activity recognition (HAR) systems by combining various machine learning techniques to address the complex nature of wearable sensor data [14,45]. Transformer models have also shown promise in enhancing HAR, leveraging their ability to capture long-range dependencies in sequential data [17,46].

The application of LLMs in HAR is a relatively new but rapidly growing area of research. Studies have highlighted the potential of LLMs to perform the zero-shot recognition of activities, significantly reducing the need for large labeled datasets [9,26]. Moreover, LLMs can integrate multimodal data, including text and sensor data, to provide more comprehensive and accurate activity recognition [24,99].

Case studies such as *PhysioLLM* and *Health-LLM* have demonstrated how LLMs can be fine-tuned for specific health-related tasks, enhancing the accuracy and relevance of predictions by incorporating contextual information [3,13]. Additionally, the use of zero-shot learning in models like *HARGPT* underscores the ability of LLMs to recognize human activities from raw sensor data without extensive training datasets [12]. Other examples like *MindShift* and *LLaSA* further illustrate the versatility of LLMs in providing mental health support and advanced human activity analysis [4,21].

### 8.2. Challenges, Trends, and Future Directions

Despite these advancements, several challenges remain. Ensuring data quality and addressing class imbalances are critical in improving HAR systems’ performance. Hybrid sampling strategies and data-efficient models, such as those utilizing Vision Transformers, have been proposed to tackle these issues [15,18]. Furthermore, the development of optimized deep learning models that can operate efficiently on resource-constrained devices is paramount [100,101].

Future research should focus on addressing these challenges while exploring new frontiers in HAR. This includes the development of more robust and scalable models, the integration of additional sensor modalities, and the application of task-specific deep learning approaches [102]. Additionally, interdisciplinary collaborations will be essential to ensure the ethical and effective deployment of these technologies [44].

Emerging trends in wearable technology, such as the integration of advanced sensors and real-time feedback mechanisms, will further enrich the data available for analysis. This, combined with the multimodal capabilities of LLMs, will lead to more comprehensive and accurate insights, benefiting fields like sports science, mental health, and workplace ergonomics [16,24,96].

Interdisciplinary research will also be crucial in addressing the ethical and privacy concerns associated with wearable sensor data. Collaborations between computer scientists, healthcare professionals, legal experts, and ethicists will help to develop frameworks that protect user privacy while enabling the effective use of LLMs for data analysis [26].

In conclusion, the integration of LLMs in wearable sensor data modeling holds significant potential to revolutionize health monitoring, activity recognition, and various other applications. By building on recent advancements and addressing the current challenges, researchers can develop more accurate, efficient, and versatile HAR systems. Continued research and collaboration across disciplines will be essential to realize the full potential of these technologies, ensuring that they are used effectively and ethically to enhance human well-being.

## Figures and Tables

**Table 1 sensors-24-05045-t001:** Types of wearable sensors.

Sensor Type	Description	Refs.
Physiological Sensors	Monitor vital signs and other physiological parameters. Examples include heart rate monitors, electrocardiograms (ECG), blood pressure monitors, and pulse oximeters.	[14]
Motion Sensors	Include accelerometers, gyroscopes, and magnetometers, used to track movement and orientation. Essential in applications like activity recognition and sports science.	[15]
Environmental Sensors	Detect environmental conditions such as temperature, humidity, and light. Often integrated into wearable devices to provide context-aware services.	[16]
Biochemical Sensors	Measure biochemical markers such as glucose levels, lactate, and electrolytes. Valuable in medical diagnostics and continuous health monitoring.	[3]
Multisensor Systems	Integrate multiple sensor types into a single device to provide comprehensive monitoring capabilities. Examples include smartwatches and fitness trackers.	[4]

**Table 2 sensors-24-05045-t002:** Nature of data generated by wearable sensors.

Data Type	Description	Refs.
Time-Series Data	Most wearable sensors produce continuous streams of time-series data, capturing dynamic changes over time. These types of data require specialized techniques for preprocessing, segmentation, and feature extraction.	[12]
Multimodal Data	Wearable devices often generate multimodal data by combining inputs from different types of sensors. For instance, a smartwatch may collect both motion and physiological data simultaneously. Integrating and synchronizing these data streams is a complex task that is essential for accurate analysis.	[4]
High-Dimensional Data	The raw data from wearable sensors can be high-dimensional, particularly when multiple sensors are used. Dimensionality reduction techniques, such as principal component analysis (PCA) and feature selection methods, are employed to manage this complexity.	[17]
Noisy and Incomplete Data	Wearable sensors are prone to generating noisy and sometimes incomplete data due to various factors like sensor malfunctions, user movement, and environmental interference. Effective data cleaning and imputation methods are critical in maintaining data quality.	[18]

**Table 4 sensors-24-05045-t004:** Capabilities and limitations of LLMs for wearable sensor data analysis.

Aspect	Description	Refs.
**Capabilities of LLMs**		
Natural Language Understanding	LLMs can comprehend complex language patterns, making them suitable for tasks such as sentiment analysis, entity recognition, and language translation. Their ability to understand context and generate relevant responses has been demonstrated in various studies, including those focusing on health data interpretation.	[3,42]
Text Generation	The text generation capabilities of LLMs are unparalleled, allowing them to produce coherent and contextually appropriate text for diverse applications. This has been leveraged in generating health-related content, educational materials, and even creative writing.	[39]
Multimodal Data Integration	LLMs can be integrated with other data modalities, such as sensor data, to provide comprehensive insights. For example, the LLaSA model combines text data with inertial measurement unit (IMU) data to enhance human activity recognition.	[4]
**Limitations of LLMs**		
Computational Requirements	Training and deploying LLMs requires substantial computational resources, which can be prohibitive for many applications. Efficient model architectures and optimization techniques are necessary to mitigate these challenges.	[15]
Data Dependency	LLMs rely heavily on large, high-quality datasets for training. The quality and diversity of the training data significantly impact the model’s performance and generalizability. Incomplete or biased data can lead to inaccurate predictions and outputs.	[12,43]
Interpretability	LLMs operate as black-box models, making it difficult to interpret their decision-making processes. This lack of transparency is a significant limitation, especially in critical applications such as healthcare, where understanding the rationale behind predictions is crucial.	[13]
Ethical Concerns	The use of LLMs raises ethical issues related to data privacy, security, and potential misuse. Ensuring compliance with data protection regulations and implementing privacy-preserving techniques are essential to address these concerns.	[44]

**Table 5 sensors-24-05045-t005:** Challenges in using LLMs for wearable sensor data.

Challenge	Description	Refs.
**General Issues Related to Using LLMs with Wearable Sensor Data**		
Data Quality and Preprocessing	Wearable sensors generate noisy, incomplete, and inconsistent data. Techniques like noise reduction, normalization, feature extraction, hybrid sampling, and data augmentation are essential for transforming raw data into a suitable format for LLMs. Effective multimodal data integration is also critical.	[1,2,48,49]
Computational Requirements	LLMs require substantial computational resources for training and fine-tuning, impacting their feasibility for real-time data analysis. Optimizing the model architectures, training algorithms, and integration with other AI techniques can help to reduce the resource requirements.	[50,51]
Interpretability and Transparency	LLMs operate as black-box models, making it difficult to understand their decision-making processes. Improving their interpretability through attention mechanisms, visualization, and explainable AI techniques is crucial for trust and accountability, especially in healthcare applications.	[52,53]
Bias and Fairness	LLMs may exhibit biases based on the demographic characteristics of the training data. Ensuring fairness involves using diverse datasets, regularly auditing model performance, and implementing bias mitigation strategies.	[43,54,55]
**Challenges Specific to Using LLMs for Sensor-Based HAR**		
Data Processing and Integration Complexities	Sensor-based HAR involves harmonizing different data types from various wearable devices. Developing robust data fusion techniques and ensuring the synchronization and alignment of multimodal data are essential.	[56,57,58,59]
Real-Time Adaptability	LLMs may struggle with low-latency requirements for real-time applications like emergency response and health monitoring. Exploring lightweight model architectures (or *tiny-LLMs*), efficient training algorithms, and edge computing can help to achieve real-time adaptability.	[60,61]
**Challenges Specific to Using LLMs for Sensor-Based Health Monitoring and Mental Health**		
Health Monitoring	The continuous tracking of physiological metrics using wearable sensors and LLMs requires accurate health predictions. Techniques like federated learning can enhance the accuracy and privacy in health monitoring systems.	[62,63,64]
Mental Health	LLMs can provide personalized mental health support but require stringent privacy protections and timely, contextually appropriate interventions. Models like EmLLMs aim to provide empathetic and personalized support.	[31,65,66,67]
**Challenges Specific to Using LLMs for Sensor-Based Behavioral Modeling in Sports and Ergonomics**		
Sports and Fitness	Wearable sensors are used to monitor performance, optimize training, and prevent injuries. Ensuring accurate and relevant insights through multimodal data integration and real-time analytics is critical for LLMs.	[32,68,69,70]
Workplace Ergonomics	Wearable sensors can improve workplace ergonomics by monitoring movements and posture, providing real-time feedback, and preventing musculoskeletal disorders. Ensuring data accuracy, personalized recommendations, and managing privacy are key challenges for LLMs.	[37,71,72,73,74]

**Table 6 sensors-24-05045-t006:** Ethical and legal issues in using LLMs for wearable sensor data.

Ethical or Legal Issue	Description	Refs.
**User Privacy Protection**		
General Privacy Concerns	Wearable devices collect extensive personal data, including health metrics, location information, and behavioral patterns. Protecting user privacy is paramount to maintain trust and ensure confidentiality.	[14,76,77,78,79]
Sensor-Specific Privacy Concerns	Privacy concerns vary with the type of sensor data (e.g., IMU, ECG, PPG). IMU data might pose a lower privacy risk compared to ECG and PPG data, which are more sensitive and related to health conditions. Stronger privacy protections are necessary for more sensitive data types processed via LLMs.	[79,80,81]
Data Handling Practices	Implementing data anonymization, minimization techniques, and transparent data handling practices is essential while using LLMs with sensor-based data.	[77,82]
**Data Security**		
General Data Security Concerns	Robust encryption methods, secure data storage solutions, and access control mechanisms are essential to safeguard wearable sensor data from unauthorized access and cyberattacks. Compliance with GDPR, HIPAA, and other regulations is necessary.	[21,83,84]
Privacy-Preserving Techniques	Techniques like federated learning and differential privacy minimize the risk of data exposure while enabling the effective use of LLMs for data analysis. These techniques keep raw data on the user’s device, sharing only aggregated model updates.	[13,85]
Sensor-Specific Data Security	Tailored security measures for different sensor types (e.g., ECG and PPG data require stronger encryption than IMU data) are necessary. Ensuring secure transmission from wearable devices to LLM-hosting infrastructure or edge computing nodes is also crucial.	[82,83,84]
**Model Bias**		
General Model Bias Concerns	LLMs are susceptible to biases in their training data, leading to unfair and discriminatory outcomes. Diverse and representative training datasets, regular audits, and bias correction techniques are essential.	[43,54,75]
Sensor-Specific Model Bias	The bias can vary with different sensor types (e.g., IMU vs. ECG) and demographic characteristics (e.g., young vs. elderly users). The continuous monitoring and adjustment of models are required to ensure fair performance across all user groups.	[86,87,88,89,90,91,92]
Transparency in Model Development	Ensuring transparency in sensor-based LLM model development and decision-making processes helps to identify and address biases effectively. Clear communication regarding how models work and how decisions are made is crucial.	[93,94,95]

## Data Availability

Data sharing is not applicable.

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
