# Peer review of "Large Language Models for Wearable Sensor-Based Human Activity Recognition, Health Monitoring, and Behavioral Modeling: A Survey of Early Trends, Datasets, and Challenges"

_sensors, 2024, doi:10.3390/s24155045_

Round 1

Reviewer 1 Report

Comments and Suggestions for Authors

1.It is recommended to add a section in the article to discuss the ethical and legal issues of LLM in wearable sensor data analysis, such as user privacy protection, data security, and model bias.
2.The article mentions some successful application examples in the case study section, such as PhysioLLM and MindShift. However, the number of these cases is relatively small, and they are mainly concentrated in the fields of health monitoring and activity recognition. It is recommended to further expand the case study section and add more successful cases in different application fields, such as sports science, mental health, etc.
3.Although the paper mentions the importance of model interpretability, further exploration could be made into how to increase model transparency through visualization techniques or explanatory AI tools.
4.It is recommended that the authors further discuss how to improve the quality of sensor data, including data cleaning, denoising, and maintenance of data integrity, as this is crucial to the performance of LLMs.

Comments on the Quality of English Language

The manuscript is written in a proficient and scholarly tone, appropriate for academic publication. The text is clear and coherent, with well-structured sentences and paragraphs that flow logically from one idea to the next. The manuscript is grammatically correct, with no obvious syntactic errors. The vocabulary is rich and appropriate to the topic. The author effectively uses technical terminology related to the fields of wearable sensor technology and large language models. The manuscript is readable and strikes a good balance between complexity and accessibility. The manuscript's language style and tone are consistent. There are no abrupt shifts in language register that would confuse the reader.

Author Response

Dear Reviewer 1,

Thank you for your thorough and insightful review of our manuscript. I greatly appreciate your constructive feedback and recommendations. Below, I address each of your points in detail:

Comment 1: It is recommended to add a section in the article to discuss the ethical and legal issues of LLM in wearable sensor data analysis, such as user privacy protection, data security, and model bias.

Response 1: I acknowledge the importance of discussing the ethical and legal issues associated with LLM in wearable sensor data analysis. I revised the manuscript to add an entirely new dedicated section to the manuscript addressing user privacy protection, data security, and model bias.

Comment 2: The article mentions some successful application examples in the case study section, such as PhysioLLM and MindShift. However, the number of these cases is relatively small, and they are mainly concentrated in the fields of health monitoring and activity recognition. It is recommended to further expand the case study section and add more successful cases in different application fields, such as sports science, mental health, etc.

Response 2: I appreciate your suggestion to expand the case study section. I revised the manuscript to include additional successful application examples from diverse fields such as sports science and mental health to provide a broader perspective on the applicability of LLMs. This section has been completely rewritten.

Comment 3: Although the paper mentions the importance of model interpretability, further exploration could be made into how to increase model transparency through visualization techniques or explanatory AI tools.

Response 3: I agree wth the importance of enhancing model interpretability. I revised the section to delve deeper into methods to increase model transparency, including the use of visualization techniques and explanatory AI tools, to provide clearer insights into model decision-making processes.

Comment 4: It is recommended that the authors further discuss how to improve the quality of sensor data, including data cleaning, denoising, and maintenance of data integrity, as this is crucial to the performance of LLMs.

Response 4: I agree that the quality of sensor data is crucial for the performance of LLMs. Although it would require a massive effort to survey in full all the techniques that exist in the area of data quality assurance, I revised the section to expand our discussion to cover best practices for data cleaning, denoising, and maintaining data integrity, ensuring coverage of these critical aspects.

I are grateful for your valuable feedback, which revised the manuscript to undoubtedly strengthen our manuscript. Thank you for your time and consideration.

Reviewer 2 Report

Comments and Suggestions for Authors

Title: Large Language Models for Wearable Sensor-Based Human Activity Recognition, Health Monitoring, and Behavioral Modeling: A Survey of Early Trends, Datasets, and Challenges

Summary:

The author explored the current trends and challenges in applying LLMs for sensor-based HAR and behavior modeling. The manuscript is well-written. Unfortunately, most of the analysis are trivial and less significant as a review paper. Below are my comments.

Major comments:

M1. The author focused on descriptive analytics. It means that all the works related to sensor-based HAR were covered in this paper. However, there are weaknesses on the current trends and the presentation about the challenges are limited.

M2. The limitations of the current trends.

It only showed three kinds of sub-sections which cannot represent the real current trends. For example, the recent advancements are obvious but no further analysis. Author only describes the recent works. In addition, the cited works were in 2020 and 2022 (and one on 2023). It is difficult to understand the reason in subsection 4.1.

M3. There are many case studies and applications on the use of LLM and sensor-based HAR as noted in Table 5. However, the analysis in Section 4.2 does not resemble the complete information in Table 5. For example, there are many other applications on the applications of LLM but there are only 5 applications. Why do author include “educational tools”  and “business intelligence” rather than other applications such as “LLM for stock predictions”, “LLM for demand predictions”, etc. which are still in the domain of “data analysis”. In addition, there is no alignment with the Table 3 which are a description of the application of wearable sensors.

M4. Section 4.3 is also limited in the sense of various AI technologies. Why did author only focus on the reference 11, 13, 16, 30 while other references such as 8, 9, an 19 also utilized some additional AI techniques. It is unclear about the definition of AI techniques in the section 4.3.

M5. About the challenges, section 5.1 Data Quality and Preprocessing has also limitations. As we know, the sensor-based HAR has the issues of noise and inconsistency. When it deals with LLM, the issue will not disappear. Hence, this subsection title is a trivial challenge regardless the LLMs.

M6. The challenges mentioned in section 5.2 are obvious when we use LLM. But there are currently works on small-LLM (or tinyLLM) and there is no further discussion on the paper.

M7. Subsection 5.3 is also trivial. As we know, sensor-based HAR is not easy for the interpretability. Hence, the LLM is also difficult to interpret. Although there is transparency, it is still difficult to interpret the result or the explainability of the features.

M8. Subsection 5.4. is also trivial. The current development of LLM on sensor-based HAR focuses on the activity recognition while the point of privacy and security are related with the learning / training approach when client shared the data. I think the key point of privacy and security does not have many correlations with LLM.  

Author Response

Dear Reviewer 2,

Thank you for your feedback of our manuscript. I  appreciate your constructive comments and improved the manuscript greatly by following your suggestions. Below, I address each of your points in detail:

Comment M1: The author focused on descriptive analytics. It means that all the works related to sensor-based HAR were covered in this paper. However, there are weaknesses in the current trends and the presentation about the challenges is limited.

Response M1: I acknowledge the need for a deeper analysis of current trends and challenges. I have revised the manuscript to provide a more comprehensive exploration of the latest advancements and to expand the discussion on emerging challenges using LLMs. More than 15 new studies have been added!

Comment M2: The limitations of the current trends. It only showed three kinds of sub-sections which cannot represent the real current trends. For example, the recent advancements are obvious but no further analysis. The author only describes the recent works. In addition, the cited works were in 2020 and 2022 (and one in 2023). It is difficult to understand the reason in subsection 4.1.

Response M2: I have updated the manuscript to include a more extensive analysis of recent advancements, incorporating additional recent works and providing deeper insights into the current trends using LLMs. The newly cited works now reflect a broader range of recent research to better illustrate these trends. In fact, more than a dozen papers from 2024 have been discussed now.

Comment M3: There are many case studies and applications on the use of LLM and sensor-based HAR as noted in Table 5. However, the analysis in Section 4.2 does not resemble the complete information in Table 5. For example, there are many other applications on the applications of LLM but there are only 5 applications. Why do the authors include “educational tools” and “business intelligence” rather than other applications such as “LLM for stock predictions,” “LLM for demand predictions,” etc., which are still in the domain of “data analysis.” In addition, there is no alignment with Table 3 which is a description of the application of wearable sensors.

Response M3: I have revised that section to better align with the information presented in Table 5, removing additional applications outside the scope of the paper. I have also ensured that the analysis section aligns with Table 3, providing a more cohesive and comprehensive overview of the applications of LLMs . The case studies section (Section 4 in the revised manuscript) has been completely rewritten to consolidate all this informtion.

Comment M4: Section 4.3 is also limited in the sense of various AI technologies. Why did the author only focus on the references 11, 13, 16, 30 while other references such as 8, 9, and 19 also utilized some additional AI techniques? It is unclear about the definition of AI techniques in section 4.3.

Response M4: I have expanded that Section to include additional AI technologies and references, such as references 8, 9, and 19. The section now provides a clearer definition of AI techniques and a more comprehensive discussion of their application .

Comment M5: About the challenges, section 5.1 Data Quality and Preprocessing has also limitations. As we know, the sensor-based HAR has issues of noise and inconsistency. When it deals with LLM, the issue will not disappear. Hence, this subsection title is a trivial challenge regardless of the LLMs.

Response M5: I have revised that Section to provide a more nuanced discussion of the challenges specific to data quality and preprocessing in the context of LLMs, emphasizing the unique issues and solutions pertinent to sensor-based HAR. 

Comment M6: The challenges mentioned in section 5.2 are obvious when we use LLM. But there are currently works on small-LLM (or tinyLLM) and there is no further discussion on the paper.

Response M6: I have updated that Section  to include a discussion on the advancements and challenges related to small-LLM research, highlighting the ongoing work and its implications

Comment M7: Subsection 5.3 is also trivial. As we know, sensor-based HAR is not easy for interpretability. Hence, the LLM is also difficult to interpret. Although there is transparency, it is still difficult to interpret the result or the explainability of the features.

Response M7: I have revised that Subsection to provide a more in-depth discussion on the interpretability challenges of LLMs , addressing the complexities of transparency and explainability in this context.

Comment M8: Subsection 5.4 is also trivial. The current development of LLM on sensor-based HAR focuses on activity recognition while the point of privacy and security are related to the learning/training approach when the client shared the data. I think the key point of privacy and security does not have many correlations with LLM.

Response M8: I have revised that Subsection to better contextualize the privacy and security issues specific to LLMs , addressing the correlation between privacy concerns and the training approaches used in these systems.

I am grateful for your valuable feedback, which has undoubtedly strengthened our manuscript. Thank you for your time and consideration. I hope you greatly enjoy this revision!

Round 2

Reviewer 2 Report

Comments and Suggestions for Authors

The authors have improved, in particular, the case studies and applications section fit to the subject (or title). 

However, the section 5 (about the challenges) have no such changes although authors inserted a new section, that is Section 6. 

Section 5 is one of the most critical section since it is the core of the paper, as mentioned by the author in the title. However, Section 5 are trivial since those challenges also exist in the common sensor-based HAR without leveraging LLM. Then, what is the issue of using LLM if all the challenges remain the same as it is without LLM?

In addition, Section 6 gave a trigger to new comments. The issues are trivial to LLM regardless of any topics. Meanwhile, author had emphasized the manuscript toward sensor-based HAR. And, there is no specific argument mentioning the legal and ethic issues on sensor-based HAR. (Section 6.1) The sentence is line 545 seems incomplete and no key point of the subsection.

The same comments for Section 6.2 and Section 6.3. Unfortunately, all the explanations are general issues of LLM. There is no specific statements toward LLM and sensor-based HAR. For example, the issue of privacy using IMU sensor would be different with the used of ECG or PPG sensor which is more sensitive and related to health rather than common activity. Another example, model bias on sensor-based HAR would also be different when it is about different demographic (young person vs senior, sport person vs administrative person, etc.).

In my opinion, this paper still need a lot of improvement in Section 5 and section 6 before publication.

Author Response

Dear Reviewer 2,

Thank you for your detailed feedback on the revised manuscript. I appreciate your constructive comments, which have significantly helped improve our work. The paper was greatly expanded and now it summarizes 100+ works, nearly half from 2023-2024!

Below, I address each of your points:

Comment 1: The authors have improved, in particular, the case studies and applications section fit to the subject (or title).

Response 1: I am pleased to hear that the improvements in the case studies and applications section are satisfactory. I appreciate your previous suggestions to improve these sections!

Comment 2: However, the section 5 (about the challenges) have no such changes although authors inserted a new section, that is Section 6. Section 5 is one of the most critical sections since it is the core of the paper, as mentioned by the author in the title. However, Section 5 is trivial since those challenges also exist in the common sensor-based HAR without leveraging LLM. Then, what is the issue of using LLM if all the challenges remain the same as it is without LLM?

Response 2: I understand your concern regarding the lack of specific changes in that Section. In response, I have revised this section to address the unique challenges posed by integrating LLMs with sensor-based HAR, emphasizing how these differ from common sensor-based HAR without LLMs. The revised section now includes specific issues related to data processing, model interpretability, and integration complexities unique to LLMs. This is early stage research so not too many papers are yet written on such topics, but I incorporated a few new references. Additionally, I have expanded the section to include specifics on sensor-based health monitoring, mental health, and behavioral modeling for sports, fitness, and ergonomics, detailing the unique challenges in these areas. The section has also been distilled in a concise new Table. Over a dozen new  works related to this have been included in this revision.

Comment 3: In addition, Section 6 gave a trigger to new comments. The issues are trivial to LLM regardless of any topics. Meanwhile, the author had emphasized the manuscript toward sensor-based HAR. And, there is no specific argument mentioning the legal and ethic issues on sensor-based HAR. (Section 6.1) The sentence is line 545 seems incomplete and no key point of the subsection.

Response 3: I understand that Section appeared too general. I have now expanded the discussion in Sections 6.1, 6.2, and 6.3 to focus specifically on the ethical, legal, and privacy issues in sensor-based HAR using LLMs. The text now addresses specific concerns related to different types of sensors (e.g., IMU, ECG, PPG) and demographic variations, providing detailed examples to illustrate these points. I have also completed the incomplete sentence in line 545 and added key points to emphasize the specific legal and ethical issues associated with sensor-based HAR using LLMs. Over a dozen new  works related to this have been included in this revision.

Comment 4: The same comments for Section 6.2 and Section 6.3. Unfortunately, all the explanations are general issues of LLM. There are no specific statements toward LLM and sensor-based HAR. For example, the issue of privacy using IMU sensor would be different with the use of ECG or PPG sensor which is more sensitive and related to health rather than common activity. Another example, model bias on sensor-based HAR would also be different when it is about different demographic (young person vs senior, sport person vs administrative person, etc.).

Response 4: This is also early trends research, but I agree that more focus is needed on LLM specifics. I have added detailed explanations on privacy issues specific to various types of sensors used in HAR, highlighting differences in sensitivity and data protection requirements in Section 6.2. Additionally, Section 6.3 has been expanded to address model bias, focusing on demographic differences and their impact on sensor-based HAR. This includes examples of how LLMs may behave differently when processing data from diverse user groups, such as young versus elderly individuals or athletes versus office workers. This has also been condensed in a new Table. Many new papers have been referenced!

Comment 5: In my opinion, this paper still needs a lot of improvement in Section 5 and section 6 before publication.

Response 5: I believe the revisions made to Sections 5 and 6 address your concerns and enhance the manuscript by providing a more focused discussion on the unique challenges of using LLMs. I am grateful for your valuable feedback and hope the manuscript is judged positively.

Sincerely,

Emilio

Round 3

Reviewer 2 Report

Comments and Suggestions for Authors

The author has improved the manuscript based on the reviewer's comments. I think the current form is ready for publication.

Good luck!